# Quality of and barriers to routine childbirth care signal functions in primary level facilities of Tigray, Northern Ethiopia: Mixed method study

Haftom Gebrehiwot Weldearegay [1] *, Alemayehu Bayray Kahsay[1], Araya Abrha Medhanyie[1], Hagos Godefay[2], Pammla Petrucka[3]

1 Mekelle University, College of Health Sciences, Mekelle, Ethiopia, 2 Tigray Region Health Bureau, Tigray, Ethiopia, 3 University of Saskatchewan, College of Nursing, Canada and Adjunct Nelson Mandela African Institute of Science and Technology, Tanzania, Canada

* haftom1224@gmail.com

**Data Availability Statement:** All relevant data are within the paper and its Supporting Information files.

## Abstract

### Background

Efforts to expand access to institutional delivery alone without quality of care do not guarantee better survival. However, little evidence documents the quality of childbirth care in Ethiopia, which limits our ability to improve quality. Therefore, this study assessed the quality of and barriers to routine childbirth care signal functions during intra-partum and immediate postpartum period.

### Methods

A sequential explanatory mixed method study was conducted among 225 skilled birth attendants who attended 876 recently delivered women in primary level facilities. A multi stage sampling procedure was used for the quantitative phase whilst purposive sampling was used for the qualitative phase. The quantitative survey recruitment occurred in July to August 2018 and in April 2019 for the qualitative key informant interview and Focus Group Discussions (FGD). A validated quantitative tool from a previous validated measurement study was used to collect quantitative data, whereas an interview guide, informed by the literature and quantitative findings, was used to collect the qualitative data. Principal component analysis and a series of univariate and multivariate linear regression analysis were used to analyze the quantitative data. For the qualitative data, verbatim review of the data was iteratively followed by content analysis and triangulation with the quantitative results.

### Results

This study showed that one out of five (20.7%, n = 181) mothers received high quality of care in primary level facilities. Primary hospitals (β = 1.27, 95% CI:0.80,1.84, p = 0.001), facilities which had staff rotation policies (β = 2.19, 95% CI:0.01,4.31, p = 0.019), maternal involvement in care decisions (β = 0.92, 95% CI:0.38,1.47, p = 0.001), facilities with

**Funding:** This study was financially supported by "Tigray KMC Project" which is funded by Bill and Melinda Gates foundation and World Health Organization; Grant Number: 201526690."The funders had no role in study design, data collection 548 and analysis, decision to publish, or preparation of the manuscript.

**Competing interests:** The authors have declared that no competing interests exist.

**Abbreviations:** KII, key informant interviews; PCA, Principal Component Analysis; QoC, Quality of Care; SARA, Service Availability and Readiness Assessment survey; SBA, Skilled Birth Attendant; WHO, World Health Organization.

maternal and newborn health quality improvement initiatives (β = 1.58, 95% CI:0.26, 3.43, p = 0.001), compassionate respectful maternity care training (β = 0.08, 95% CI: 0.07,0.88, p = 0.021), client flow for delivery (β = 0.19, 95% CI:-0.34, -0.04, p = 0.012), mentorship (β = 0.02, 95% CI:0.01, 0.78, p = 0.049), and providers' satisfaction (β = 0.16, 95% CI:0.03, 0.29, p = 0.013) were predictors of quality of care. This is complemented by qualitative research findings that poor quality of care during delivery and immediate postpartum related to: work related burnout, gap between providers' skill and knowledge, lack of enabling working environment, poor motivation scheme and issues related to retention, poor providers caring behavior, unable translate training into practice, mismatch between number of provider and facility client flow for delivery, and in availability of essential medicine and supplies.

## Conclusions

There is poor quality of childbirth care in primary level facilities of Tigray. Primary hospitals, facilities with staff rotation, maternal and newborn health quality improvement initiatives, maternal involvement in care decisions, training on compassionate respectful maternity care, mentorship, and high provider satisfaction were found to have significantly increased quality of care. However, client flow for delivery service is negatively associated with quality of care. Efforts must be made to improve the quality of care through catchment-based mentorship to increase providers' level of adherence to good practices and standards. More attention and thoughtful strategies are required to minimize providers' work-related burnout.

## Introduction

The period around childbirth and the first 24 hours postpartum remains a perilous time for both mother and newborn despite global efforts and improvements in mortality over the past two decades [1–2]. Previously, adverse outcomes were thought to result primarily from delivery occurring outside of health facilities and from lack of access to skilled care [3–4]. However, a recent literature review showed more women in low income countries are delivering in facilities but this shift has not been consistently linked to mortality reduction nor guaranteed that appropriate interventions are rendered during the intra-partum and immediate postpartum period in settings which may result in low quality of care (QoC) [5–7]. Strategies such as the Janani Suraksha Yojana (JSY) program, a large conditional cash transfer programs in India, have increased rates of facility-based childbirth without significantly decreasing maternal and neonatal mortality [6, 8]. This finding implies improving the QoC provided during facility-based childbirth is key to decreasing maternal and neonatal mortality and complications [9,10].

Measuring QoC for mothers and newborns is multi-dimensional and conceptually complex, reflecting both the provision (in terms of structure, process and outcomes) and experiences of care. According to Donabedian, the measurement of process (how care is delivered) is nearly equivalent to the measurement of QoC because process encompasses all acts of health care delivery [11,12]. Furthermore, one approach to simplify this complexity is to focus on the content of care received as defined by the routine processes that are recommended to occur during a contact between health care user and provider. It has been argued that these processes provide a measurable and meaningful indication of QoC since they describe the potential for health-gain in given clinical scenario. In the context of maternal and newborn health care it is

possible to define content of care by drawing on the global recommendations for minimum packages of routine care that should be made available to all women and newborns during intra-partum and immediate postpartum care [13,14].

The use of evidenced-based practices for routine care and management of complications is the key to achievement of high QoC [15]. Receiving a QoC is also a universal human right and should apply to all women everywhere. All women and newborn have a right to access a QoC that enables a positive childbirth experience that includes respect and dignity, a companion of choice, clear communication with maternity staff, pain relief strategies, mobility in labor, and position of choice [16,17]. Recent evidence indicated that poor quality of care at primary health care facilities, the first point of contact with the health system, not only jeopardizes the health of mothers and newborns, but erodes trust, resulting in bypassing health facilities and potentially puts the entire healthcare system and population at risk [18].

Despite mothers attending facilities for deliveries, there is mismatch between maternal and newborn services provided and demand for QoC due to limited skilled providers, knowledge deficits, lack of confidence, environment lacking consistent structured education at pre-service and in service, and lack of access to evidence based current information [6, 19]. In addition, motivation of health workers to translate knowledge into action is an important influencer of quality of care [20,21]. In Ethiopia a great deal of attention and investment of resources has been directed towards basic emergency obstetrics care training. However, this initiative alone has not translated into improved QoC. This discrepancy has given rise to the quality gap in maternal and newborn care implying that the content of care provided in health facilities is often of insufficient quality to have a major impact on avoidable deaths and complications. Therefore, in recent years a more focused attention to QoC has been raised [6, 22]. Few studies have been conducted on QoC of routine childbirth care functions and almost all of them did not use a validated instrument for measuring the QoC, making comparisons difficult [23]. Consequently, identifying the possible improvement pathways of the QoC during this critical period could have a substantial impact on maternal and newborn survival and inform hospital managers, professionals, researchers and policy makers about where improvements may be focused to enhance effectiveness of health services. Therefore, assessing the QoC provided routinely for uncomplicated childbirth and identifying barriers to QoC in primary health facilities of Tigray, Northern Ethiopia is essential.

## Materials and methods

A facility based cross-sectional explanatory sequential mixed method study was conducted among recently delivered mothers in primary health care facilities of Tigray regional state, Northern Ethiopia. Tigray is the northern most region of Ethiopia with an estimated total population of 5,247,005 with 21.2% of the population living in urban areas and 50.3% being female [24]. The maternal and new born health care services in the region are provided mainly by emergency obstetric surgeons and obstetricians at hospitals and health centers by midwives, nurses, and health officers. The service is given free of charge in all public health facilities on a seven day per week 24 hours a day basis. As of 2016, in Tigray Region, there were 2 comprehensive specialized hospitals, 15 general hospitals, 23 primary hospitals, 214 health centers, and 718 health posts [25, 26].

### Quantitative phase

a sample size calculation for the quantitative study component among recently delivered mothers was determined by a single population proportion with 95% confidence interval, margin of error (d) of 5% and taking 54.06% prevalence (P) of overall quality of delivery care in Arbaminch,

south Ethiopia public health facilities [27]; design effect of 2 and adding 10% for non-response rate. A total of 881 mothers received routine intra-partum and immediate postpartum care signal functions from 40 primary level care facilities. Additionally, a total of 225 skilled birth attendants (SBAs) working in the study facilities at the time of data collection were included.

A multi-stage sampling procedure was adopted to select the districts and primary level health facilities from each district. In the first stage, three of the seven zones were selected randomly. In the second stage, nine of the 22 districts were chosen and 6 primary hospitals were randomly selected. Thereafter, all health centers with their respective catchment primary level hospitals were included with the total sample size being distributed over each of the health facilities proportionate to their sample considering average number of deliveries per facility per month. All SBAs in the study were enrolled. Finally, all eligible recently delivered women were chosen by a systematic random method until the required sample size was achieved. A referred mother requiring care at a higher level facility for further management and/or delivered by cesarean section were excluded from this study. Client exit interview tracer indicators for routine childbirth care signal functions (care that should be provided for all mothers and newborns) utilized self-administered questionnaires, facility inventories, and interviews of providers to collect the quantitative data. A 40-item knowledge tests, as well as satisfaction of health workers and facility readiness surveys (i.e., availability of infrastructure; essential medications and commodities; guidelines; staff) were conducted. Tracer indicators for facility readiness were used from the WHO Service Availability and Readiness Assessment (SARA) list, previously reported indices [28]. Twelve data collectors and three supervisors worked as data collection teams. Data collectors had previous research experience and trained for two days. Data for this quantitative study was collected between July to August 2018.

## Qualitative phase

We developed semi-structured questionnaires to conduct key informant interviews (KII) and Focus Group Discussions (FGD). Participants for FGDs and KIIs were selected purposively. The key informant participants were medical directors from each of the primary level health facilities, woreda, and regional health bureau maternal and child health experts and unit head of maternity wards. We assumed that these KII could better inform us of barriers to provide childbirth care than other health workers. A total of twelve KIs were conducted. Probing questions were used for a better understanding where necessary. Each participant was interviewed individually at his/her place of work, with interview duration ranging from 20 to 35 minutes by a team of trained data collectors. After training, three researchers from the College of Health Sciences at Mekelle University and Tigray Health Research Institute conducted qualitative data collection from the 9th to 29th of April 2019. The semi-structured interview guide used can be found in **S1 Appendix**.

After the interviews, three FGD were held for SBAs working at intra-partum and immediate postpartum care ranging from 60 to 90 minutes. One interviewer and note taker were involved. Skilled birth attendants with clinical work experience of six months and below were excluded from the qualitative study. All interviews and discussions were audio recorded, then transcribed verbatim in Tigrigna (the local dialect) by two independent investigators. A third investigator checked the consistency of the transcripts and verified the transcripts by listening to the tapes again. They were subsequently translated into English prior to analysis.

## Variables and measurements

The primary outcome investigated was quality of routine childbirth and immediate postpartum care. It was measured as a continuous variable constructed as a composite variable from the total of 32 standards of quality process of care indicators. The routine intra-partum and immediate

postpartum care signal functions used in this study are grounded in validated indicators in the Tigray regional state context. Detail of the measurement and validated tool findings is found in the recent article submitted for publication [29]. Principal component analysis (PCA), the most common technique of creating a single or composite quality index, which is a variable reduction method to obtain a smaller set of uncorrelated variables from a large list of correlated variables, was used. Each component is a linear combination of the observed variables optimally weighted to account for the maximum amount of variance [30]. Therefore, quality measures reflect the minimum standards of routine intra-partum and immediate postpartum care, irrespective of the type of health facilities where the delivery service is performed. According to the PCA, QoC was defined as a binary variable of "low" to "high" on a continuous scale from 0 to 100. If a mother's review received 75% and above, it was termed as high QoC, and otherwise received low QoC. Details of the PCA tool for measuring QoC is found in **S2 Appendix.**

The providers' satisfaction variable was classified as "satisfied" (providers scored 75[th] percentile and above), whereas below the 75th percentile was considered "not satisfied"; facility readiness was categorized as adequately ready at the 75[th] percentile and above and below was considered inadequately ready). Details of the PCA tool for measuring providers' satisfaction is found in the **S3 Appendix.**

Knowledge of providers on intra-partum and immediate postpartum care signal functions was determined using a set of 31 multiple choice questions and 9 true or false questions. Each correct answer was valued at one point, and a wrong answer attracted no points. Questions that were not answered were treated as wrong answers. Ultimately, participants were evaluated out of 100, and grouped as either sufficient knowledge (median or higher) or insufficient knowledge (less than median value).

## Data management and analysis

**Quantitative analysis.**   First, we entered the data in to EPI data, cleaned and analyzed it using SPSS™ version 21 software. Descriptive statistics were used to summarize the characteristics of delivered mothers, facilities, and providers. Characteristics of the study population were presented with mean and standard deviation for variables with normal distribution. The normality of distribution of quantitative variables was tested by Kolmogorov–Smirnov test. We used linear regression analysis to assess the association between quality of care and explanatory variables. Simple linear regression analyses were conducted and those independent variables with p value of $\leq 0.25$ were considered for multiple linear regression with the forward likelihood ratio method. Finally, statistical significance was considered if $p < 0.05$.

Furthermore, an index score of PCA was done after checking the suitability of the data. The correlation coefficient was set at a cut-off point of 0.4 or above. The Kaiser-Meyer-Oklin value, which was used to assess sampling adequacy, was set at a cut-off point of 0.5 [30], while the Bartlett's test of sphericity was used to support the factorability of the correlation matrix. Furthermore, a scree plot tests and eigenvalue of over 1.0, which represents the total variance explained by a factor, were used to inspect the plotting of each eigenvalue of the factors to find a point at which the shape of the curve changes direction and becomes horizontal. All factors above the break in the plot and/or with eigenvalues over 1.0 were retained for further analysis. Lastly, further analysis was done using the Vari-max method to minimize the number of variables with high loadings on each factor.

## Qualitative analysis

Two researchers independently reviewed the audio recorded comments line- by- line and then agreed on a set of codes; broadly categorized into those related to the quantitative checklist

and codes for other emerging issues. Both researchers then jointly coded all the open-ended comments. In cases where disagreements arose between researchers, further discussion took place until consensus was achieved. The data analysis was carried out in three stages. First, familiarization involving reading and re-reading the transcripts to aid understanding of the data. Second, organizing and coding the data. The coding was determined based on the quantitative results, to aid understanding how the quantitative findings were manifest. The coding was done using Atlas ti™7.5 software. Third, data from each code point were reviewed and summarized to reduce the number of words without losing the content or context of the text and to ensure contents were internally consistent. Then content analysis and triangulation of data were done through a continuous back and forth interpretation of findings.

### Ethics approval and consent

The study protocol was approved by the Institutional Research Review Board of Mekelle University's College of Health Sciences and Community Services Ethical Review Committee (ERC 1436/2018). Permission was obtained from all relevant authorities in the Tigray Regional Health Bureau and health facilities. Informed consent was obtained from all participants prior to enrollment in the study. Parental or legal guardian consent was obtained for participants who were under 18 years of age. Data collection was conducted confidentially while data was de-identified and de-linked with storage in a secure location.

## Results

### Socio-demographic characteristics of mothers

A total of 876 mothers who delivered in the primary health care facilities were included in the study with a response rate of 99.43%. Above half of the mothers (n = 465, 53.1%) were within the age group 25–34 years and ranged from 17 to 45 years (mean age = 28.9, SD = 6.1). Two thirds of the mothers (587; 67.0%) lived in rural setting and 610 (69.6%) were housewives. More than three-quarters (n = 789, 89.5%) of the participants were married, 91% (n = 798) belonged to the Orthodox Christian religion, and nearly 40% (n = 343) had no formal education. More than six out of ten mothers 566 (64.6%) walked greater than 30 minutes to the nearest health facility [Table 1].

### Reproductive history of mothers

Seven hundred ninety-seven (91.0%) mothers had antenatal care (ANC) visits for their current pregnancy with 30.6% (n = 242) having four or more ANC visits. Seven out of ten mothers (n = 561) were gave birth in the same facility where they received ANC follow up. Around two thirds (n = 535) of women had birth preparedness and complication readiness plan. Over one third (n = 379) of the participants had between two and four pregnancies. While 510 (58.2%) mothers had between 2 to 5 children ever born. Around two out of ten (161; 18.6%) mothers had a history of abortion, while 11.4% (n = 100) had a history of stillbirth. With respect to allowing partners to enter the delivery room, about two thirds (n = 594) of women had allowed their partners to enter and receive support in the delivery room. Around three fourths of mothers (n = 664) had been involved in decision making for the type of care they received during childbirth and soon after. With respect to obstetrical complications, one hundred seven (12.2%) of mothers had faced an obstetrics complication. Of those, pregnancy induced hypertension was the most common obstetrical complication (n = 38, 35.5%) [Table 2].

Table 1. Socio-demographic characteristics of mothers in Northern Ethiopia, 2019 (N = 876).

| Variable | Number | Percentage |
|---|---|---|
| **Age in years** | | |
| 15–24 | 227 | 25.9 |
| 25–34 | 465 | 53.1 |
| 35 and above | 184 | 21.0 |
| **Residence** | | |
| Rural | 587 | 67.0 |
| Urban | 289 | 33.0 |
| **Mother's occupation** | | |
| Housewife | 610 | 69.6 |
| Employed | 158 | 18.0 |
| Daily worker | 108 | 12.4 |
| **Marital status** | | |
| Married | 784 | 89.5 |
| Single/never married | 43 | 4.9 |
| Divorced/widowed/separated | 49 | 5.6 |
| **Religion** | | |
| Orthodox Christian | 798 | 91.1 |
| Muslim | 59 | 6.7 |
| Others* | 19 | 2.2 |
| **Mother's education** | | |
| No formal education | 343 | 39.2 |
| Elementary school | 282 | 32.2 |
| Secondary school and above | 251 | 28.6 |
| **Estimated walking time to the nearest health facility** | | |
| 30 minute and below | 310 | 35.4 |
| Greater than 30 minutes | 566 | 64.6 |

*Other religions = Catholic and Protestants

## Socio-demographic characteristics of skilled birth attendants

The average age of SBAs was 29.7 years (SD ± 7.0) with a range of 21 to 58 years. The majority of SBAs (55.1%) were between 25 and 35 years. Health providers at delivery were predominantly staff midwives (52.4%).

Over half of SBAs (51.6%) providing intra-partum care were registered diploma holders and around two thirds (n = 148) of the providers attended regular program education.

Most of the SBAs (52.0%) had worked in the obstetrics unit providing intra-partum and immediate postpartum care for 2 to 5 years [**Table 3**].

## Barriers of skilled birth attendants to QoC

Table 4 shows that about three fourths of the SBAs [74.7% (n = 168)] were dissatisfied with their existing job. Six out of ten of the SBAs (n = 137) reported ever attending a formal basic emergency obstetrics and newborn care training, followed by neonatal resuscitation or helping babies breathe (42.7%) during the past two years. One hundred and four (46.2%) of providers were knowledgeable on basic obstetrics care practices. The average SBAs knowledge score on routine intra-partum and immediate postpartum care functions was 22.61(±5.4) with the range scored from 9 to 37 out of a total 40 item questions. **S4 Appendix** shows the basic

**Table 2. Reproductive history of mothers in Northern Ethiopia, 2019 (n = 876).**

| Variables | Number | Percentage |
|---|---|---|
| ANC visit for the current pregnancy | | |
| Yes | 797 | 91.0 |
| No | 79 | 9.0 |
| Number of ANC visits | | |
| 1 | 331 | 41.9 |
| 2–3 | 217 | 27.5 |
| 4 and above | 242 | 30.6 |
| Place where ANC was received | | |
| Health Center | 487 | 61.1 |
| Hospital | 283 | 35.5 |
| Health Post | 27 | 3.4 |
| Does your last ANC visit was in this facility? | | |
| Yes | 561 | 70.4 |
| No | 236 | 29.6 |
| Birth preparedness and complication readiness (BPCR) | | |
| Yes | 535 | 61.1 |
| No | 341 | 38.9 |
| Length of labor | | |
| <12 hours | 758 | 86.5 |
| ≥12 hours | 118 | 13.5 |
| Mode of Delivery | | |
| Spontaneous vaginal delivery (SVD) | 760 | 86.8 |
| Instrument delivery | 116 | 13.2 |
| How long do women generally stay at the facility following a normal delivery? | | |
| <6 hours | 508 | 58.0 |
| 6–24 hours | 282 | 32.2 |
| >24 hours and above | 86 | 9.8 |
| Number of pregnancies/Gravidity | | |
| 1 Pregnancy | 113 | 20.1 |
| 2–4 Pregnancies | 379 | 67.6 |
| 5 and above pregnancies | 69 | 12.3 |
| Number of deliveries/Parity | | |
| Primipara (1 delivery) | 230 | 26.3 |
| Multipara (2-5deliveries) | 510 | 58.2 |
| Grand multipara (5 and above deliveries) | 136 | 15.5 |
| History of abortion | | |
| Yes | 163 | 18.6 |
| No | 713 | 81.4 |
| History of stillbirth | | |
| Yes | 100 | 11.4 |
| No | 776 | 88.6 |
| Mothers allowed their partner to enter to the delivery room | | |
| Yes | 594 | 67.8 |
| No | 282 | 32.2 |
| Maternal involvement in care decisions | | |
| Yes | 664 | 75.8 |
| No | 212 | 24.2 |

(*Continued*)

**Table 2.** (Continued)

| Variables | Number | Percentage |
|---|---|---|
| Complication(s) during the current pregnancy | | |
| Yes | 107 | 12.2 |
| No | 769 | 87.8 |
| Type of complication(s) (n = 107) | | |
| Hemorrhage | 24 | 22.4 |
| Pregnancy Induced Hypertension | 38 | 35.5 |
| Infection | 24 | 22.4 |
| Others* | 21 | 19.6 |

Others complication*: Anemia, tear, delay of expulsion of placenta and head ache

emergency obstetrics standard questionnaire/tool used for assessing the knowledge of SBAs. The qualitative result revealed that perceived lack of legal protection in terms of medical indemnity insurance, poor motivation, or benefit packages (risk allowance, low salary and lack of opportunity for further education etc..), lack of an enabling environment, poor leadership and governance, lack of capacity building mechanisms, and mismatch of number of providers and facility capacities to conduct deliveries were the main barriers of satisfaction of providers.

**Table 3.** SBAs' background characteristics working at obstetrics in primary health facilities of Tigray, Northern Ethiopia, 2019 (N = 225).

| Variable | Number | Percentage |
|---|---|---|
| Age of provider in completed years | | |
| $\leq 25$ | 62 | 27.6 |
| 25–35 | 124 | 55.1 |
| > 35 | 39 | 17.3 |
| Marital status | | |
| Married | 119 | 52.9 |
| Divorced | 15 | 6.7 |
| Single | 91 | 40.4 |
| Provider work experience in years | | |
| Less than 5 years | 117 | 52.0 |
| 5 years and above | 108 | 48.0 |
| Sex of provider | | |
| Male | 75 | 33.3 |
| Female | 150 | 66.7 |
| Highest level of education | | |
| Diploma | 116 | 51.6 |
| Degree and above | 109 | 48.4 |
| Educational program attended | | |
| Generic | 148 | 65.8 |
| Upgrade regular | 48 | 21.3 |
| Upgrade in-service | 29 | 12.9 |
| Professional cadre | | |
| Midwife | 118 | 52.4 |
| Nurse | 69 | 30.7 |
| Health officer and MD | 38 | 16.9 |

**Table 4. Readiness of SBAs working in obstetrics at primary health facilities of Tigray, Northern Ethiopia, (N = 225).**

| Variable | Number | Percentage |
|---|---|---|
| In the last 2 years received basic emergency obstetrics training | | |
| Yes | 137 | 60.9 |
| No | 88 | 39.1 |
| Neonatal resuscitation/helping babies to breathe training | | |
| Yes | 96 | 42.7 |
| No | 129 | 57.3 |
| Compassionate respectful care training | | |
| Yes | 80 | 35.6 |
| No | 145 | 64.4 |
| Quality improvement training | | |
| Yes | 42 | 18.7 |
| No | 183 | 81.3 |
| Legal issues fear to make decision in your daily basis | | |
| Yes | 50 | 22.2 |
| No | 175 | 77.8 |
| In the last 6 months, received clinical mentorship | | |
| Yes | 108 | 48.0 |
| No | 117 | 52.0 |
| Support supervision in the last 6 months | | |
| Yes | 154 | 68.4 |
| No | 71 | 31.6 |
| Had regular case presentation in your team/facility | | |
| Yes | 97 | 43.1 |
| No | 128 | 56.9 |
| Recommendation to improve quality of obstetrics care | | |
| Pre service/in-service training | 89 | 39.6 |
| Catchment based mentorship | 115 | 51.1 |
| Supportive supervision | 21 | 9.3 |
| Having challenge in providing intra-partum, and immediate postpartum care? | | |
| Yes | 79 | 35.1 |
| No | 146 | 64.9 |
| Postnatal women checked and discharged by senior staff of the facility | | |
| Yes | 70 | 31.1 |
| No | 155 | 68.9 |
| Satisfaction of skilled birth attendants | | |
| Satisfied | 57 | 25.3 |
| Not satisfied | 168 | 74.7 |
| Knowledge of skilled birth attendants | | |
| Appropriate knowledge | 121 | 53.8 |
| Inappropriate knowledge | 104 | 46.2 |
| Weekly delivery cases per individual (mean ±SD) | | 4.04 ±2.70 |

The main concern of midwives was lack of authority to make decisions. For example, initiating necessary early referral of laboring mothers in case of complication prediction is often delayed due to systemic processes. An experienced midwife said *"we don't have enough power to decide on timely and appropriate referral of laboring mother for higher and better*

*management. Em. . .Yea. . . referral issues are expected to handle by health officers in the health centers. However, this leads unnecessarily delay until the responsible person called up and arrange referral slip which in turn again provoking further complication and death"* (FGD participant, degree midwife).

In addition, this study specified that fear of legal issues was an important barrier to satisfaction of SBAs. One health center director described *"Since the safety of mother and newborn currently is very momentous for all providers and policy makers, many decisions are made based on panic legal issues, mixing up of politics and health care. Everyone in the health system is terribly afraid of litigation"* (KII participant, Health center director).

## Facility characteristics

Eighty five percent of the participating health facilities were health centers. Eight out of ten health facilities did not have a regular staff rotation policy. 62.5% (n = 25) of the primary health care level facilities did not introduce any maternal and newborn health quality improvement initiative. With respect to facility readiness, more than half 21(52.55%) of the facilities were assessed as adequately ready [**Table 5**].

## Quality of process of routine childbirth care signal functions

This study showed that two out of ten (n = 181) mothers received high quality of routine child birth care signal functions in Tigray region, northern Ethiopia (**Fig 1**).

## Barriers for quality of routine childbirth care signal functions

Results of the fully adjusted regression analysis (Table 6) reveal that facility type, regular staff rotation, facility-based maternal and newborn health quality improvement initiatives, provider

**Table 5. Facility characteristics in Northern Ethiopia, 2019 (n = 40).**

| Variables | Number | Percentage |
|---|---|---|
| Facility Type | | |
| Health Center | 34 | 85.0 |
| Primary Hospital | 6 | 15.0 |
| Facility has maternal, perinatal/neonatal death surveillance & responding (MPNDSR) | | |
| Yes | 29 | 72.5 |
| No | 11 | 27.5 |
| Facility had regular staff rotation policy | | |
| Yes | 8 | 20.0 |
| No | 32 | 80.0 |
| Maternal Newborn Health collect the data regularly | | |
| Yes | 30 | 75.0 |
| No | 10 | 25.0 |
| Facility has mobile data internet access | | |
| Yes | 37 | 92.5 |
| No | 3 | 7.5 |
| Maternal Newborn Health quality improvement initiative | | |
| Yes | 15 | 37.5 |
| No | 25 | 62.5 |
| Facility Readiness | | |
| Inadequately ready | 19 | 47.5 |
| Adequately ready | 21 | 52.5 |

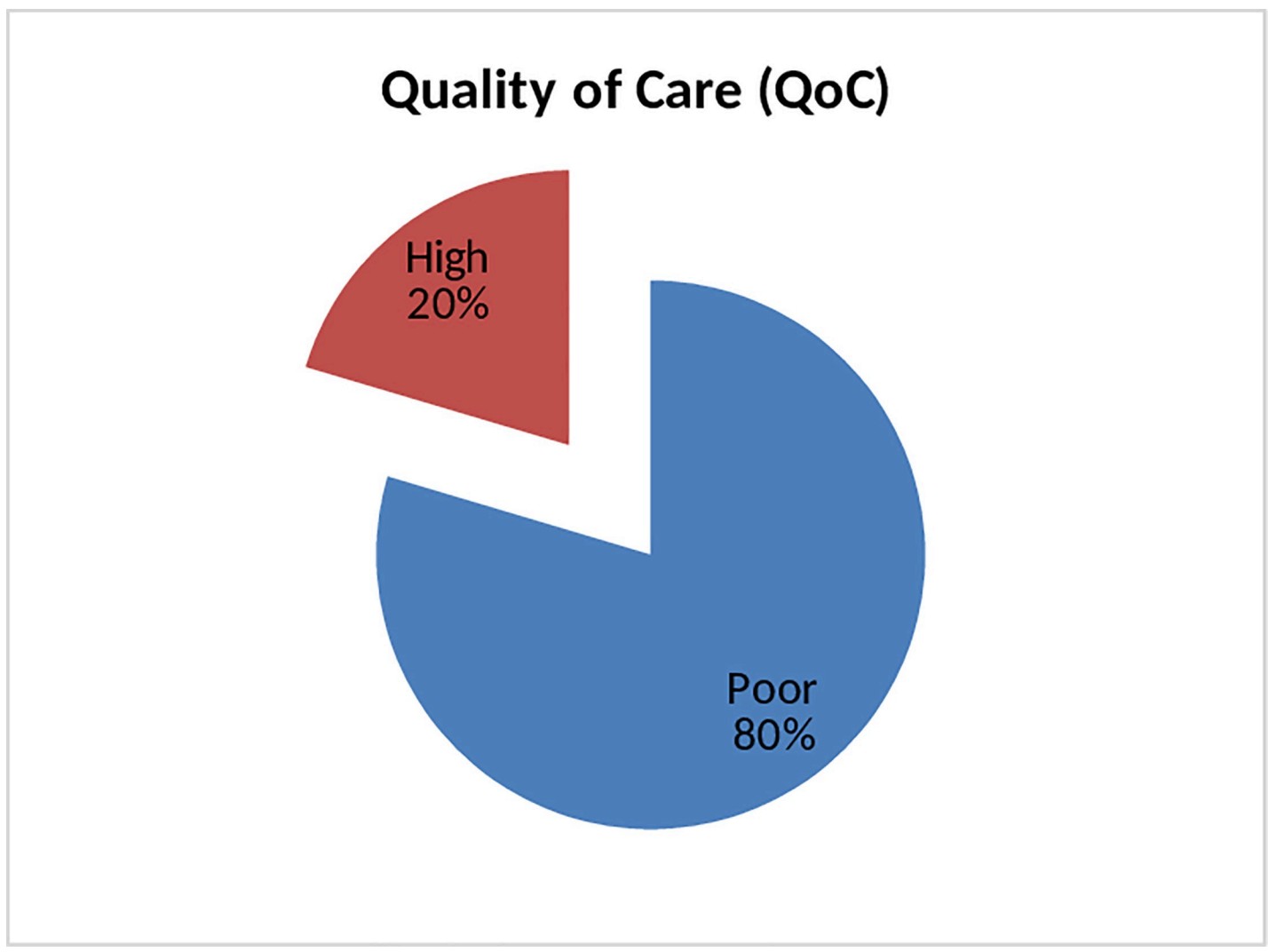

**Fig 1. Quality of routine childbirth care signal functions.**

training on compassionate respectful maternity care, client flow for delivery, mentorship opportunities, and evidence of providers' satisfaction significantly impacted on quality of routine childbirth care signal functions.

Having one unit increase in maternal and newborn health care quality improvement initiatives in the facility, the quality of routine childbirth care signal functions provision increased significantly (β = 1.58, 95% CI: 0.26, 3.43; p = 0.001).

Every one-unit change in level of facility type (i.e., from health center to primary hospital) resulted in 1.27 increase in provision of QoC (β = 1.27, 95% CI: 0.80, 1.84; p = 0.001).

Due to change in receiving compassionate respectful maternity care training of providers in the last two years, the provision of high quality of child birth care signal functions increased significantly (β = 0.08, 95% CI: 0.07,0.88; p = 0.021). Facilities that had a staff rotation policy in place for maternal and newborn health units (more than one rotation a year) were 2.19 times more likely to provide QoC compared to facilities without staff rotation policies (β = 2.19, 95% CI: 0.01, 4.31; p = 0.019).

**Table 6. Linear regression analysis of predictors on quality of routine childbirth care signal functions in primary health facilities of Tigray, Northern Ethiopia, 2019.**

| Variables | Univariate Analysis | | | Multivariate Analysis | | |
|---|---|---|---|---|---|---|
| | B | 95%CI | P-value | β adj. | 95%CI | P |
| Maternal residence | | | | | | |
| Rural | Ref | | | | | |
| Urban | 0.15 | (-0.34,0.64) | 0.552 | | | |
| Maternal education | | | | | | |
| No formal education | Ref | | | | | |
| Elementary school | -0.17 | (-0.67,0.33) | 0.501 | | | |
| Secondary school and above | -0.20 | (-0.71,0.31) | 0.443 | | | |
| **Mothers involved in decision of their care** | | | | | | |
| No | Ref | | | **Ref** | | |
| Yes | 1.17 | (0.64,1.71) | 0.000 | **0.92** | **(0.38,1.47)** | **0.001** |
| Estimated walking time to the nearest health facility | | | | | | |
| 30 minute and below | Ref | | | | | |
| Greater than 30 minutes | -0.11 | (-0.59,0.38) | 0.664 | | | |
| ANC visit for current pregnancy | | | | | | |
| No | Ref | | | Ref | | |
| Yes | 0.88 | (0.07,1.68) | 0.034 | 0.62 | (-0.19,1.42) | 0.134 |
| Birth preparedness and complication readiness (BPCR) | | | | | | |
| No | Ref | | | Ref | | |
| Yes | 0.29 | (-0.19,0.76) | 0.233 | -0.05 | (-0.53,0.43) | 0.839 |
| Length of labor in hours | -0.13 | (-0.20,0.15) | 0.601 | | | |
| Mode of Delivery | | | | | | |
| Spontaneous (SVD) | Ref | | | Ref | | |
| Instrument Delivery | -0.72 | (-1.41,-0.04) | 0.038 | -0.64 | (-1.31,0.04) | 0.064 |
| Time women stay at the facility following a normal delivery | | | | | | |
| < 6hours | 0.06 | (-0.41,0.53) | 0.800 | | | |
| 6–24 hours | -0.21 | (-0.70,0.29) | 0.414 | | | |
| >24 hours | Ref | | | | | |
| Parity/deliveries | -0.08 | (-0.17,0.01) | 0.075 | -0.02 | (-0.14,0.09) | 0.694 |
| Number of pregnancies/Gravidity | -0.16 | (-0.28,-0.04) | 0.007 | -0.06 | (-0.23,0.11) | 0.460 |
| History of abortion | | | | | | |
| No | Ref | | | Ref | | |
| Yes | -0.63 | (-1.11,-0.16) | 0.009 | -0.46 | (-0.96,0.05) | 0.075 |
| History of stillbirth | | | | | | |
| No | Ref | | | Ref | | |
| Yes | -0.65 | (-1.38,0.08) | 0.081 | -0.44 | (-1.21,0.33) | 0.261 |
| Allow your partner to enter to the delivery room | | | | | | |
| No | Ref | | | | | |
| Yes | 0.02 | (-0.47,0.52) | 0.926 | | | |
| **Facility Type** | | | | | | |
| Health Center | Ref | | | Ref | | |
| Primary Hospital | **-1.63** | **(-4.11,0.85)** | **0.192** | **1.27** | **(0.80,1.84)** | **0.001** |
| Facility has MPNDSR | | | | | | |
| No | Ref | | | | | |
| Yes | 1.07 | (-0.92,3.07) | 0.284 | | | |
| **Facility had regular staff rotation policy** | | | | | | |

*(Continued)*

**Table 6.** (Continued)

| Variables | Univariate Analysis | | | Multivariate Analysis | | |
|---|---|---|---|---|---|---|
| | B | 95%CI | P-value | β adj. | 95%CI | P |
| No | Ref | | | Ref | | |
| Yes | **1.66** | **(0.54,3.85)** | **0.135** | **2.19** | **(0.01,4.31)** | **0.019** |
| Facility has mobile data internet access | | | | | | |
| No | Ref | | | | | |
| Yes | -1.75 | (-0.14,1.94) | 0.303 | | | |
| **Maternal and newborn health quality improvement initiative** | | | | | | |
| No | Ref | | | Ref | | |
| Yes | **0.72** | **(-2.58,1.14)** | **0.437** | **1.58** | **(0.26,3.43)** | **0.001** |
| Facility readiness | -0.24 | (-0.55,0.11) | 0.178 | 1.23 | (0.81,2.04) | 0.247 |
| Age of the provider | -0.02 | (-0.08,0.03) | 0.407 | | | |
| Provider work experience | 0.02 | (-0.08,0.04) | 0.500 | | | |
| Sex of provider | | | | | | |
| Male | Ref | | | | | |
| Female | 0.09 | (-0.72,0.91) | 0.833 | | | |
| SBAs' highest level of education | | | | | | |
| Diploma | Ref | | | Ref | | |
| Degree and above | 0.66 | (-1.42,0.90) | 0.088 | -0.69 | (-1.56,0.17) | 0.115 |
| Educational program attended | | | | | | |
| Generic | Ref | | | Ref | | |
| Upgrade | -0.71 | (-1.33,-0.09) | 0.024 | 0.003 | (-0.82,0.81) | 0.995 |
| Professional cadre | | | | | | |
| Midwife | 0.54 | (-0.14,1.21) | 0.122 | 0.03 | (-1.18,1.23) | 0.963 |
| Nurse | 0.49 | (-0.37,1.35) | 0.263 | -0.49 | (-1.79,0.82) | 0.463 |
| Health officer and MD | Ref | | | Ref | | |
| In the last 2 years received basic emergency obstetrics training | | | | | | |
| No | Ref | | | | | |
| Yes | 0.40 | (-0.38,1.18) | 0.315 | | | |
| **Compassionate respectful maternity care training** | | | | | | |
| No | Ref | | | Ref | | |
| Yes | **-0.06** | **(-0.85,0.74)** | **0.893** | **0.08** | **(0.07,0.88)** | **0.021** |
| Quality improvement training | | | | | | |
| No | Ref | | | Ref | | |
| Yes | -0.39 | (-1.37,0.59) | 0.434 | | | |
| **Client flow for delivery** | **-0.21** | **(-0.35,-0.07)** | **0.003** | **-0.19** | **(-0.34,-0.04)** | **0.012** |
| Legal issues fear to make decision in your daily basis care | | | | | | |
| No | Ref | | | | | |
| Yes | -0.24 | (-1.22,0.74) | 0.625 | | | |
| **Receive a clinical mentorship** | | | | | | |
| No | Ref | | | Ref | | |
| Yes | **-0.08** | **(-0.84,0.69)** | **0.842** | **0.02** | **(0.01,0.78)** | **0.049** |
| Receive support supervision | | | | | | |
| No | Ref | | | Ref | | |
| Yes | 0.86 | (0.04,1.67) | 0.040 | -1.23 | (-2.43,1.98) | 0.768 |
| Had regular case presentation (morning session) | | | | | | |
| No | Ref | | | | | |
| Yes | 0.37 | (-0.40,1.14) | 0.347 | | | |

(Continued)

**Table 6.** (Continued)

| Variables | Univariate Analysis | | | Multivariate Analysis | | |
|---|---|---|---|---|---|---|
| | B | 95%CI | P-value | β adj. | 95%CI | P |
| Having challenge in providing intra-partum, and immediate postpartum care | | | | | | |
| No | Ref | | | | | |
| Yes | 0.37 | (-0.43,1.17) | 0.365 | | | |
| Motivation of HCPs | 0.16 | (-0.30,0.62) | 0.493 | | | |
| **Satisfaction of HCPs** | **0.18** | **(0.05,0.31)** | **0.005** | **0.16** | **(0.03,0.29)** | **0.013** |
| Knowledge of HCPs | -0.02 | (-0.09,0.05) | 0.630 | | | |

NB: β adj is adjusted β; P is P-value, CI confidence interval

Every unit increase in receiving clinical mentorship lead to in 0.02 increases in the provision of QoC. For every one unit increase in involving of mothers to their care decisions, resulted in 0.92 increases in received high QoC during their routine childbirth care functions. Similarly, one unit increase in client flow for delivery of the provider resulted in 0.19 decrease in the provision of QoC (β = 0.19, 95% CI: -0.34, -0.04; p = 0.012). For every one unit increase in provider job satisfaction, the provision of QoC increased significantly (β = 0.16, 95% CI: 0.03, 0.29; p = 0.013).

According to the qualitative research findings, work related burnout, gap between providers' skills and knowledge, being fear of litigation, poor motivation schemes and issues related to their retention, shortage of SBAs mainly midwives, lack of authority to make decisions, unable to translate training into practice and unavailability of adequate medications and necessary equipment were important reasons for poor quality of care during childbirth and immediate postpartum care. These issues are reflected in a series of quotations highlighted herein:

*While poor motivation and satisfaction pressures due to low salaries and allowance were found to greatly affect the care SBAs delivered, they were not the only principal barriers. Fear of law suit, shortage of human resource, unhygienic infrastructure and inadequate availability of medicine and supplies impact up on quality practice (KII participant, unit head of maternity ward).*

*. . . heavy work load leads providers' burn out resulting from insufficient and can lead to poor performance, inappropriate behavior and attitude. In addition, limited proper capacity building devices and incompetent SBAs remain serious barriers to provide quality of maternity care services in Tigray (KII participant, Woreda MCH expert).*

*Despite much training conducted so far, whatever their role is significant in improving knowledge and skill of providers, but should have to be supplemented through coaching and mentoring which in turn increased their level of confidence in delivering services and*

*enabled them to increase adherence to good practice and*

*standards.*" (FGD participant, diploma midwife)

*Almost all SBAs have a fear of litigation in provision of maternal and*

*newborn care services. Thus, have a legal protection of SBAs in terms*

*of medical indemnity insurance is important to apply their highest*

*level potential in reducing unnecessary referral and averting maternal and*

*newborn death.*(KII participant, hospital medical director)

Another identified reason was poor communication between provider and parturient women and capacity of providers to adhere to standards (example: Partograph and active management of third stage of labor). A senior midwife said *"adherence to standard guidelines and an institutionalization of World Health organization safe childbirth checklist is very poor"* (FGD participant, degree midwife).

## Discussion

This study examined the quality of and barriers to routine intra-partum and immediate post-partum care functions among primary level health care facilities in Tigray regional state in northern Ethiopia using a mixed method approach. We found low QoC (only one out of five mothers received high QoC) overall. Primary hospitals, facilities which promote staff rotation, facilities having maternal and newborn health quality improvement initiatives, involvement of mothers in care decisions, training on compassionate respectful maternity care, client flow for delivery service, mentorship and providers' satisfaction were identified as significant predictors of QoC. This finding is complemented through the qualitative results that emphasizes work related burnout, gap between providers' skills and knowledge, lack of enabling working environment (fear of litigation and lack of authority to make decisions), poor motivation scheme and issues related to retention, poor provider caring behavior, and unable translate training into practice were important reasons for poor QoC during the delivery and immediate postpartum care period. This is lower compared to study reports done in some Sub-Saharan Africa countries [31–33]. However, this finding is slightly higher compared to a study conducted in Tanzania (14%) [34]. These variations may be due to differences in measuring standards between studies, timing of data collection, study participants included, type of health facilities, or a combination of these factors. This implies that SBAs are not adhering to standard guidelines, neglecting services to mothers and newborns, missing the most essential basic interventions, or have poor caring behaviors and skills to provide the routine childbirth care functions.

We observed a positive significant association between QoC of childbirth and staff rotation policy within maternal and newborn continuity of care units. The providers who had staff rotation policy appear best suited for provision of QoC without feeling of professional fatigue. This result is supported by study done in china [35]. This suggests that staff rotation may also reduce staff burnout and allow providers to improve their provision of quality of maternity care services. In the current study, mothers who involved to their care decisions during childbirth were more likely to receive high quality of care; this study result is consistent with studies done in Eretria [36] and in accordance with the Lancet's global quality agenda [37]. Clearly the provision of adequate information and time for women to make informed decisions about their care and treatment in partnership with their healthcare professionals is a pivotal component of standards of maternity care services, which, in turn, increased trust and confidence in

receiving continuous support, and ease of communication. One possible reason could be the majority of health providers were not attending to what women want or expect and giving priority to conducting the procedures before asking permission in advance, which is an emerging concern in Ethiopia. This align with a previous study which emphasized, with increasing service utilization, the importance of optimal interpersonal communication and involvement of mothers in decision making is likely to be a crucial dimension to maintain or increase the quality of health services [38].

Type of facility was significantly affected the quality of childbirth care provision. Those providers who are in primary hospitals were more likely to provide QoC compared to those at health centers. This result is consistent with study done Swedish [39] and Nigeria [40]. The explanation for this might lie in hospitals hosting senior staff, with many of them potentially affiliated with teaching institutions. The healthcare workers in such institutions will continually get a chance of updating their knowledge during ward round, bedsides with students, and via a series of seminars that are usually organized as a protocol of the institution. This implies that experience sharing of health centers from their catchment hospitals through regular mentoring program could enable the providers to provide high QoC services. Furthermore, our findings revealed that clinical mentorship leads to increase in the provision of high QoC of routine childbirth care signal functions.

We also found that factors at the provider level, rather than the facility level, seem to influence quality of routine childbirth care. This finding is comparable with evidence from a systematic review that outlined how several individual providers' factors (incompetency and negative behaviors and inadequate number of staffs) affect QoC [41]. To provide QoC for laboring women and newborns in health-care facilities primarily requires appropriate staffing with high competency and motivation and with the minimum availability of essential physical resources [28].

This result further showed that delivery caseload is negatively associated with quality of care. Similarly, providers who had high numbers of deliveries were found to be less likely to provide QoC [42], which was further corroborated in studies done in Malawi [43] and other sub-Saharan African countries [21, 23] This can account for significant inequity in workloads for staff in different facilities and indeed in different units within the same facility. Participants in this study reported significant increases in the number of deliveries, mirroring the sharp increase in preference for facility-based delivery; however, there has not been a parallel increase in the number of staff to attend these women.

Other findings showed that health care providers trained on compassionate respectful maternity care were significantly associated with provision of high QoC. This result was in line with studies done in Malawi [44] and Ghana [45]. This might be due to lack of exposure to caring behaviors and poor communication between providers and clients, lack of regular updates in training, and minimal certification processes before graduation contributing to poor competencies in maternity and newborn care practice. Thus, simulation based routine and continual compassionate respectful maternity care training need to be organized for improvement of providers' caring behaviors to minimize negative behaviors and increase their competency to adhere to standards.

Similarly, in this study, the provision of QoC has a direct relationship with providers' job satisfaction. This is congruent with the studies done in countries of Afghanistan [46] and Pakistan [47]. This indicates that, although there has been increased interest among researchers and policymakers in identifying and implementing effective solutions to address SBAs directed motivation strategies in remote and rural areas in recent years, the current evidence available to guide policymakers on adoption and adaptation of specific retention strategies remains quite limited [48]. Thus, more attention needs to be given to develop interventions and

strategies that directly enhance provider satisfaction and retention mechanisms in various contexts to improve QoC. Moreover, almost all SBAs have a fear of litigation in provision of maternal and newborn care services which is strongly suggestive of the urgent need to have a legal protection of SBAs through medical indemnity insurance in Tigray region.

Although inputs or facility readiness should serve as a foundation for high-quality care, our study did not suggest the existence of inputs necessary for providing better care within the existing infrastructure. This finding is similar to prior studies [5, 9, 49] which found that increased availability of inputs for delivery care are poorly correlated with provision of evidence-based care or explained an insignificant fraction of increased QoC delivered to women and newborns in need. This finding implies that, unless providers translate knowledge/evidence into practice, having well-equipped facilities might not often guarantee to provide high quality care and vice versa.

As a limitation of this study, we could not relate the outcome variable which is QoC during the routine childbirth and immediate postpartum period with those near miss mothers and deaths since many mothers in primary level facilities were immediately referred to higher facilities for better management. In case where such complications arise, the provider might use different standards in managing the complications other than for the normal birth. Therefore, to minimize this variability, we excluded the mothers with complications and only mothers with normal birth were interviewed. Hence, it is necessary to consider those limitations while interpreting the findings and conducting further research in general and in tertiary hospitals among those outlier mothers would be more plausible. However, this study had much strength. Quality of care was assessed using context based validated indicators which provide a more detailed picture of the state of QoC in the process of childbirth. In addition, women's exit interviews, shortly following delivery, but prior to facility discharge might have shortened recall time and yielded more clarity in the results.

## Conclusions

There is poor QoC during the intra-partum and immediate postpartum period in primary level facilities of Tigray, Northern Ethiopia. Primary hospitals, facilities which promote staff rotation, facilities having maternal and newborn health quality improvement initiatives, maternal inclusion in decisions related to their care, training on compassionate respectful maternity care, mentorship and providers' satisfaction were linked with significant increases in QoC. However, client flow for delivery is negatively associated with QoC. This finding was complemented by the second phase (i.e, the qualitative approach) that revealed work related burnout, gap between providers' skills and knowledge, lack of enabling working environments (fear of litigation), poor motivation scheme and issues related to retention, poor provider caring behaviors, lack of translations of training into practice, mismatch between the number of provider and facility client flow for delivery and lack of essential medicines and supplies were major bottlenecks in the provision of timely and quality obstetric care, which has a significant impact on maternal and neonatal outcomes.

Therefore, efforts must be made to improve the QoC through experience sharing of health facilities within their respective catchments, and have a legal protection of SBAs in terms of medical indemnity insurance. More attention and thoughtful strategies that match providers to workload, coupled with targeted efforts to support providers' satisfaction and health-care worker performance and retention, are necessary to mitigate the effects of working in this context and to improve the quality of obstetric care.

Initiating or providing regular catchment-based mentoring and adopting quality improvement initiatives for skilled providers are essential in order to increase adherence to good

practices and standards. Furthermore, having staff rotation within maternal newborn units (more than one rotation a year) may minimize work related burnout which, in turn, leads to improved QoC in our context.

## Supporting information

**S1 Appendix. The qualitative research guide both FGD and key informant interview to Regional head/Woreda/Director of hospital, health center and senior SBAs mainly Midwives.** (DOCX)

**S2 Appendix. Detail measurement tool of potential QoC indicators during child birth and immediate postpartum period through the Principal Component Analysis (PCA).** (DOCX)

**S3 Appendix. Detail measurement tool of providers satisfaction through the Principal Component Analysis (PCA).** (DOCX)

**S4 Appendix. Basic emergency obstetrics related standard knowledge questionnaire assessment tool for SBAs working on intra-partum & immediate postpartum care.** (DOCX)

**S1 File.** (SAV)

## Acknowledgments

Our heartfelt thanks go to Mekelle University and Tigray regional health bureau for the follow up and technical support. Our appreciation and thank is also forwarded to the research assistants and study participants for their genuine support and participation.

## Author Contributions

**Conceptualization:** Haftom Gebrehiwot Weldearegay, Alemayehu Bayray Kahsay, Araya Abrha Medhanyie, Hagos Godefay.

**Data curation:** Haftom Gebrehiwot Weldearegay.

**Formal analysis:** Haftom Gebrehiwot Weldearegay, Alemayehu Bayray Kahsay.

**Funding acquisition:** Hagos Godefay.

**Investigation:** Haftom Gebrehiwot Weldearegay.

**Methodology:** Haftom Gebrehiwot Weldearegay, Alemayehu Bayray Kahsay.

**Project administration:** Hagos Godefay.

**Software:** Haftom Gebrehiwot Weldearegay.

**Supervision:** Alemayehu Bayray Kahsay, Araya Abrha Medhanyie, Hagos Godefay, Pammla Petrucka.

**Validation:** Araya Abrha Medhanyie.

**Visualization:** Pammla Petrucka.

**Writing – original draft:** Haftom Gebrehiwot Weldearegay, Araya Abrha Medhanyie.

**Writing – review & editing:** Alemayehu Bayray Kahsay, Pammla Petrucka.

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
