## [Decision Letter · Decision Letter 0]

25 Mar 2020

PONE-D-19-31921

Quality of and Barriers to Routine Childbirth Care Signal Functions in Primary Level Facilities of Tigray, Northern Ethiopia: Mixed Method Study

PLOS ONE

Dear Haftom Gebrehiwot Weldearegay 

Thank you for submitting your manuscript to PLOS ONE. After careful consideration, we feel that it has merit but does not fully meet PLOS ONE’s publication criteria as it currently stands. Therefore, we invite you to submit a revised version of the manuscript that addresses the points raised during the review process.

We would appreciate receiving your revised manuscript by 27 April, 2020. To enhance the reproducibility of your results, we recommend that if applicable you deposit your laboratory protocols in protocols.io, where a protocol can be assigned its own identifier (DOI) such that it can be cited independently in the future. For instructions see: http://journals.plos.org/plosone/s/submission-guidelines#loc-laboratory-protocols

We look forward to receiving your revised manuscript.

Kind regards,

Sharon Mary Brownie

Academic Editor

PLOS ONE

Additional Editor Comments (if provided):

Reviewers have identified a number of areas to improve your manuscript. Please consider and respond to each and every recommendation.

Journal Requirements:

2. We note that you have reported significance probabilities of 0 in places. Since p=0 is not strictly possible, please correct this to a more appropriate limit, eg 'p<0.0001'.

Please provide an amended Funding Statement that declares *all* the funding or sources of support received during this specific study (whether external or internal to your organization) as detailed online in our guide for authors at http://journals.plos.org/plosone/s/submit-now.  Please state what role the funders took in the study.  If any authors received a salary from any of your funders, please state which authors and which funder. If the funders had no role, please state: "The funders had no role in study design, data collection and analysis, decision to publish, or preparation of the manuscript."

5. Your ethics statement must appear in the Methods section of your manuscript. If your ethics statement is written in any section besides the Methods, please move it to the Methods section and delete it from any other section. Please also ensure that your ethics statement is included in your manuscript, as the ethics section of your online submission will not be published alongside your manuscript.

Reviewers' comments:

Reviewer's Responses to Questions

**Comments to the Author**

1. Is the manuscript technically sound, and do the data support the conclusions?

Reviewer #1: Yes

Reviewer #2: Yes

2. Has the statistical analysis been performed appropriately and rigorously? 

Reviewer #1: Yes

Reviewer #2: Yes

3. Have the authors made all data underlying the findings in their manuscript fully available?

Reviewer #1: Yes

Reviewer #2: Yes

4. Is the manuscript presented in an intelligible fashion and written in standard English?

Reviewer #1: Yes

Reviewer #2: Yes

5. Review Comments to the Author

Reviewer #1: Overall the paper is good. The authors have successfully addressed the main objective of the paper. They have applied different data collection techniques and triangulate those very appropriately. Ethical procedure was followed rightly for protocol and obtaining written consent during data collection. It was easy to follow as the introduction, method, result and conclusion section were well organized and supported every section. However, some of the section; particularly in the methodology section should be clearer.

Major issues:

In the methodology section, overall line numbers 118-173 for qualitative part should rewrite; for example, data collection methods, participants’ category and data organization should write consistently and detail.

Specifically, line numbers 118-119, 155-158, please check the sentences and make consistence when you write about the study participants. In the first sentence you have written about applying qualitative approach among Skilled Birth Attendant and in the sentences lines 155-158 you have mentioned about 12 key informants. So, there has two types of participants and you have used two type data collection tools for qualitative data collection, but you have written separately which is confusing.

In the line numbers 229-233 Please write more detail about qualitative data preparation, coding, theme selection and content analysis. Did you use any software for data coding or analysis? If so, please mention.

Minor issues:

i.In the abstract; line number 30, you have mentioned focus group , it’s actually Focus Group Discussion (FGD) a specific method name. Please write Focus Group Discussion.

ii. Line no 31-32, you have used validated tools. Would you please explain how the tools were validated. If it is through field test prior to use for filed data collection you should mention it here.

iii. Did you prepare any verbatim transcript for interpretation? Verbatim interpretation words are not clear. Line number 34-35

iv. The Abstract needs some modification, in result section while stating the predictors of quality of care, in bracket including the p values along with coefficient value and confidence intervals would portray the predictors more clearly [line 37 to line 42].

v. Line 49-59 conclusion section can be a bit shorter focusing the main points in few sentences.

vi. Line 79 Please use capital letter before the writer name, Donabedian

vii. Line 84 please write quality of care instead of only quality

viii. Line 121-122, 49.7% population are male? Why did you use this data, not clear.

ix. Title: Line number 275, as you have described barrier of providers in this section. Could you please include the word into your title? It might give idea about the section to the reader when they will see the title of the paragraph.

x.You have given reference for some quotation; for instance Line no 300. Reference interview 6 . Instead of interview 6, can you please write the interview of skilled birth attendants or a key informant interview.

xi. In Result section, while interpreting Table 6 i.e. Barriers for Quality of Routine Childbirth Care Signal Functions, interpretation could be written differently, instead of stating the results in same way, for some variables you could interpret as, due to change in independent variable the predicted variable increased significantly or otherwise [line 320 to line 337]. It is not mandatory to mention the adjusted coefficient value each time. Moreover, while you are stating the 95% confidence interval and p-value, you should also include the beta value i.e. co-efficient value. I think this section needs some tightening.

xii.Line number 370 write mixed method approach

xiii. In line number 373 you can write over patient flow for delivery instead of delivery load.

Other comments: In addition, please check formatting and some spelling while you go through the paper.

Reviewer #2: This is an interesting manuscript. However, there are some issues that need to be addressed

1. There are a lot of grammatical errors in the manuscript and I recommend revising the entire manuscript, see examples below:

- A total of twelve interviewee were conducted (page 14, lines 157-158).

- We were used principal component analysis (page 15, line 184)

- In addition, PCA was done to create an index score. Prior to perform PCA, the suitability of data was assessed (page 16, lines 219-220).

2. The authors should consider presenting the results of the qualitative data analysis separate from that of the quantitative data, and in doing do should provide additional information of the participants and not just FGD participant or Interviewee 6. For example highlighting their role/occupation.

3. In addition, when presenting the results of the qualitative interviews, the authors should separate the responses of participants, for example on page 20, the authors lumped together the responses for interviewee 6 and 8.

4. The authors should consider providing a Good Reporting of A Mixed Methods Study (GRAMMS) checklist.

6. PLOS authors have the option to publish the peer review history of their article (what does this mean?). If published, this will include your full peer review and any attached files.

Reviewer #1: No

Reviewer #2: No

---

## [Author Response · Author response to Decision Letter 0]

1 May 2020

Editor-in-Chief

PLOS ONE

Subject: Response to reviewers

 Re: PONE-D-19-31921 Version 1: April 10, 2020

Dear Dr. Brownie, Academic Editor and Respected Reviewers,

We are grateful for your consideration of our manuscript to publish at PLOS ONE. We thank the reviewers for their careful review and comments that have substantially improved our manuscript. 

Responses to the editor’s and reviewers’ comments are provided below and corresponding edits have been made to the manuscript in track changes. 

Please let us know if you have any further questions or suggestions.

Sincerely, 

Haftom Gebrehiwot (PI)

---

## [Decision Letter · Decision Letter 1]

26 May 2020

Quality of and barriers to routine childbirth care signal functions in primary level facilities of Tigray, Northern Ethiopia: Mixed method study

PONE-D-19-31921R1

Dear Dr. Haftom Gebrehiwot Weldearegay,

We are pleased to inform you that your manuscript has been judged scientifically suitable for publication and will be formally accepted for publication once it complies with all outstanding technical requirements.

With kind regards,

Sharon Mary Brownie

Academic Editor

PLOS ONE

Additional Editor Comments (optional):

Reviewers comments have been addressed.

Reviewers' comments:

Reviewer's Responses to Questions

**Comments to the Author**

1. If the authors have adequately addressed your comments raised in a previous round of review and you feel that this manuscript is now acceptable for publication, you may indicate that here to bypass the “Comments to the Author” section, enter your conflict of interest statement in the “Confidential to Editor” section, and submit your "Accept" recommendation.

Reviewer #1: All comments have been addressed

Reviewer #2: All comments have been addressed

2. Is the manuscript technically sound, and do the data support the conclusions?

Reviewer #1: Yes

Reviewer #2: Yes

3. Has the statistical analysis been performed appropriately and rigorously? 

Reviewer #1: Yes

Reviewer #2: Yes

4. Have the authors made all data underlying the findings in their manuscript fully available?

Reviewer #1: Yes

Reviewer #2: Yes

5. Is the manuscript presented in an intelligible fashion and written in standard English?

Reviewer #1: Yes

Reviewer #2: Yes

6. Review Comments to the Author

Reviewer #1: Overall the manuscript is well. The data collection process, analysis has been written logically which supported the main ideas of the manuscript. I made some major and minor comments in my first review that had been attached here; I found that all the comments have been addressed very nicely. I accept this paper without further any comments.

Reviewer #2: Thank you for taking time to address the reviewers comments. The manuscript has been strengthened further.

7. PLOS authors have the option to publish the peer review history of their article (what does this mean?). If published, this will include your full peer review and any attached files.

Reviewer #1: Yes: RASHIDA AKTER

Reviewer #2: No

---

## [Editor Report · Acceptance letter]

4 Jun 2020

PONE-D-19-31921R1 

Quality of and barriers to routine childbirth care signal functions in primary level facilities of Tigray, Northern Ethiopia: Mixed method study 

Dear Dr. Weldearegay:

I'm pleased to inform you that your manuscript has been deemed suitable for publication in PLOS ONE. Congratulations! Your manuscript is now with our production department. 

Kind regards, 

on behalf of

Professor Sharon Mary Brownie 

Academic Editor

PLOS ONE